# Ground-truth encoding of self-motion in the primate cerebellar nodulus and uvula

Robyn L. Mildren ✉ & Kathleen E. Cullen

Accurate internal estimates of self-motion and orientation relative to gravity are fundamental for stabilizing gaze, controlling posture, and navigating through dynamic environments. Prevailing theories propose that the cerebellar nodulus and uvula (NU) employ internal models to suppress sensory input arising from predictable, self-generated motion. However, this assumption has never been directly tested. Here, we recorded NU Purkinje cell activity in rhesus monkeys during active and passive head movements. We found neurons responsive to passive translations remained equally sensitive to self-generated movements, encoding net head motion in space irrespective of its source. Furthermore, external perturbation did not influence these ground-truth encoding. When active head motion was blocked, Purkinje cell activity remained unchanged – demonstrating a lack of efference copy integration. During active tilts, NU neurons encoded both dynamic motion and static orientation relative to gravity. These findings challenge the internal model hypothesis and establish the NU as a ground-truth, context-invariant estimator of self-motion, supporting stable behavior in dynamic environments.

To successfully navigate the world, the brain must maintain accurate estimates of self-motion and spatial orientation relative to gravity to stabilize gaze and posture. These estimates are computed by integrating multimodal sensory information – including vestibular (i.e., semicircular canal and otolith), visual, somatosensory, and motor-related signals[1–4]. During everyday behaviors, much of this sensory feedback arises from active exploration, where distinguishing between inputs generated by our own actions and those that are externally driven or unexpected is essential for perceptual stability and accurate motor control (reviewed in Brooks and Cullen[5]). Recent evidence suggests cerebellar circuits in the anterior vermis use efference copies of motor commands to build forward internal models that predict the sensory consequences of behavior[6]. This computation underlies the suppression of expected reafferent signals downstream in the cerebellar output nuclei[7–10], resulting in selective encoding of sensory prediction errors generated by unexpected perturbations and motor errors – a framework often assumed to apply across the cerebellum. However, beyond signaling unexpected events to drive reflex responses for balance, the brain must also maintain a stable, context-invariant estimate of self-orientation and motion to support functions such as spatial awareness, arousal, and autonomic regulation. How the brain meets these competing demands – and where such a veridical "ground-truth" representation is computed – remains unknown.

One candidate for generating such a context-invariant estimate is the nodulus and uvula (NU) – lobules X and IX of the posterior cerebellar vermis – which integrate inputs from the semicircular canals and otolith organs to encode self-motion in a gravity-centered reference frame[11,12]. The NU is commonly assumed to filter out self-generated motion via internal forward models, selectively encoding externally generated perturbations[12]. However, several anatomical and physiological features suggest an alternative role. Notably, the NU receives robust monosynaptic input from vestibular afferents that encode head motion with high fidelity regardless of behavioral context[13–16]. These afferents maintain consistent tuning during both passive and active movements, including voluntary head turns and natural locomotion, raising the possibility that the NU functions as an inertial "ground-truth" estimator rather than a predictive filter. Moreover, NU Purkinje cells project to regions of the fastigial nucleus implicated in spatial awareness, arousal, and autonomic control – systems that require a veridical, context-independent representation of motion and

Department of Biomedical Engineering, Johns Hopkins University, Baltimore, Baltimore, USA. ✉e-mail: Kathleen.Cullen@jhu.edu

orientation[17–20]. Collectively, this anatomical and functional evidence positions the NU as a strong candidate for providing veridical self-motion estimates essential for both motor and non-motor functions.

Here, we directly tested whether and how NU Purkinje cells – the output neurons of the cerebellar cortex – encode actively generated head movements. We found that neurons sensitive to passive translations maintained their sensitivity during comparable active translations. Furthermore, when active and passive motion were combined, NU Purkinje cells encoded the total head movement in space without distinguishing the source. During perturbations in which neck motor commands were issued but head motion was prevented, Purkinje cell activity remained unchanged, indicating no integration of efference copy signals. For active head tilts, which require integrating otolith and semicircular canal input to determine the motion and sustained head position relative to gravity, Purkinje cells faithfully encoded both the dynamic motion and static tilt. Taken together, our findings overturn the prevailing view that the NU primarily implements predictive suppression of reafference, demonstrating instead that it provides a robust, context-independent code for self-motion and spatial orientation. This veridical encoding, unique within the vestibular cerebellar vermis, positions the NU as an essential hub for both motor and non-motor functions – serving as a stable, gravity-centered ground truth against which internal predictions can be evaluated during voluntary movement.

## Results

We recorded high-density extracellular neural activity from Purkinje cells in the NU of two alert rhesus monkeys. Purkinje cells included in this study were sensitive to passive vestibular stimulation and insensitive to eye movement. We first applied anterior or posterior directed passive whole-body translations to assess each Purkinje cell's sensitivity to vestibular stimulation (Fig. 1A, first column; passive whole-body condition, and Supplementary Fig. 1). Using the same motion profile, we then assessed sensitivity to neck proprioceptive stimulation by translating their body while their head was held stationary in space (Fig. 1A, second column; passive body under head condition). In our population, we found that the majority (95.5%, $N = 74$) of Purkinje cells that were sensitive to vestibular stimulation were also sensitive to neck proprioceptive stimulation. Responses of an example Purkinje cell to passive vestibular and proprioceptive stimulation can be seen in Supplementary Fig. 2A, and the distribution of responses across the population are shown in Supplementary Fig. 2B. Overall, NU Purkinje cell responses to vestibular and neck proprioceptive stimulation were comparable to previous recent findings[21].

### NU Purkinje cells provide ground truth signals about self-motion

Vestibular feedback received by the brain during daily life often results from our own voluntary movements. However, prior neurophysiological studies of primate NU Purkinje cells have focused exclusively on how these neurons encode vestibular signals elicited by passively applied, externally generated motion. Accordingly, we next recorded the responses of NU Purkinje cells during actively generated head movements and compared them to responses evoked by experimentally applied passive head movements matched in kinematics. To directly compare sensory encoding across conditions, we recorded NU Purkinje cell activity while monkeys actively translated their heads relative to their bodies for a reward along the anteroposterior axis (Fig. 1A, fourth column; active head on body condition). We then applied passive head-on-body translations with matched kinematics to generate comparable vestibular and neck proprioceptive stimulation in the absence of self-generated motion (Fig. 1A, third column; passive head on body condition).

The results for two example Purkinje cells – one whose firing was more in phase with head velocity (Cell 1) and another aligned with head acceleration (Cell 2) – are shown in Fig. 1B. Both cells exhibited consistent and robust modulation across repeated trials of passive head translation (Fig. 1B, left panel). To characterize their temporal response dynamics, we applied a least-squares dynamic linear regression model incorporating three kinematic terms (head velocity, acceleration, and jerk) to fit firing rates during passive head movements (solid blue line superimposed on the gray mean firing rate). With this model, the average variances accounted for (VAF) were comparable in the passive and active condition ($0.43 \pm 0.18$ and $0.41 \pm 0.17$, respectively ($p = 0.5$)). We then used this passive condition–derived model to predict how each Purkinje cell would respond during active head movements, assuming its kinematic sensitivity remained unchanged (dashed blue line). As shown in the right panels of Fig. 1B, both example cells exhibited robust modulation during active head on body translations, evident in both the mean firing rate (gray shaded region) and trial-by-trial responses. Importantly, for each neuron, the passive condition–derived model prediction during active motion closely matched the prediction generated from passive responses (solid red line), indicating that both cells maintained similarly strong modulation during active and passive head translations.

The observations from these two example Purkinje cells were representative of the broader population and are summarized in Fig. 1C for the subset of cells that remained well-isolated across both active and passive head-on-body translation conditions ($N = 95$). Across the population, NU Purkinje cells exhibited comparable firing rate sensitivities to motion in both active and passive conditions, as illustrated by the overlapping distributions of response gains (Fig. 1C, histograms). The mean sensitivity (dotted vertical lines) did not differ significantly between active and passive conditions (preferred: active $19.8 \pm 14.9$ vs. passive $19.8 \pm 13.0$ sp/s/m/s², $p = 0.6021$), indicating that, for most cells, self-generated motion did not suppress or enhance encoding of head kinematics. This pattern held true for responses in both the preferred (Fig. 1C) and non-preferred (Supplementary Fig. 3A) directions of movement (non-preferred: active $21.2 \pm 19.5$ vs. passive $22.2 \pm 19.2$ sp/s/m/s², $p = 0.6847$). These results suggest that, at the population level, NU Purkinje cells maintain consistent encoding of self-motion regardless of whether movements are externally applied or actively generated.

While the above analysis emphasized the similarity in average sensitivity across the population (Fig. 1C), an equally important feature of NU Purkinje cell responses is their marked heterogeneity in temporal tuning. Importantly, this diversity is preserved during self-generated motion. As shown in the heat plots of Fig. 2A, individual neurons exhibit a wide range of firing phases, collectively tiling the full cycle of self-motion. This richness in response dynamics, evident under passive conditions, remains intact during active head-on-body translations. Moreover, the sensitivity and phase of individual neurons were strongly correlated between active and passive conditions in both the preferred (Fig. 2B; sensitivity R² = 0.7728, phase R² = 0.8646, $p$-values < 0.0001) and non-preferred (Supplementary Fig. 3B; sensitivity R² = 0.4439, phase R² = 0.8722, $p$-values < 0.0001) directions. Neurons with positive sensitivities preferred posterior motion, while those with negative sensitivities preferred anterior motion (Fig. 2B), and the magnitudes of these sensitivities clustered symmetrically across the two directions. Correspondingly, phases tended to cluster around a similar phase lead or lag relative to acceleration. Polar plots illustrate that neurons maintain their tuning characteristics across contexts. Although mean sensitivities did not differ, the preserved heterogeneity in temporal dynamics underscores the stability and richness of population coding in the NU during both externally applied and self-generated motion.

To further probe the diversity of temporal tuning across movement contexts, we next classified individual Purkinje cell responses based on their firing rate dynamics during active and passive head-on-body motion. Using a model-based decomposition of each neuron's

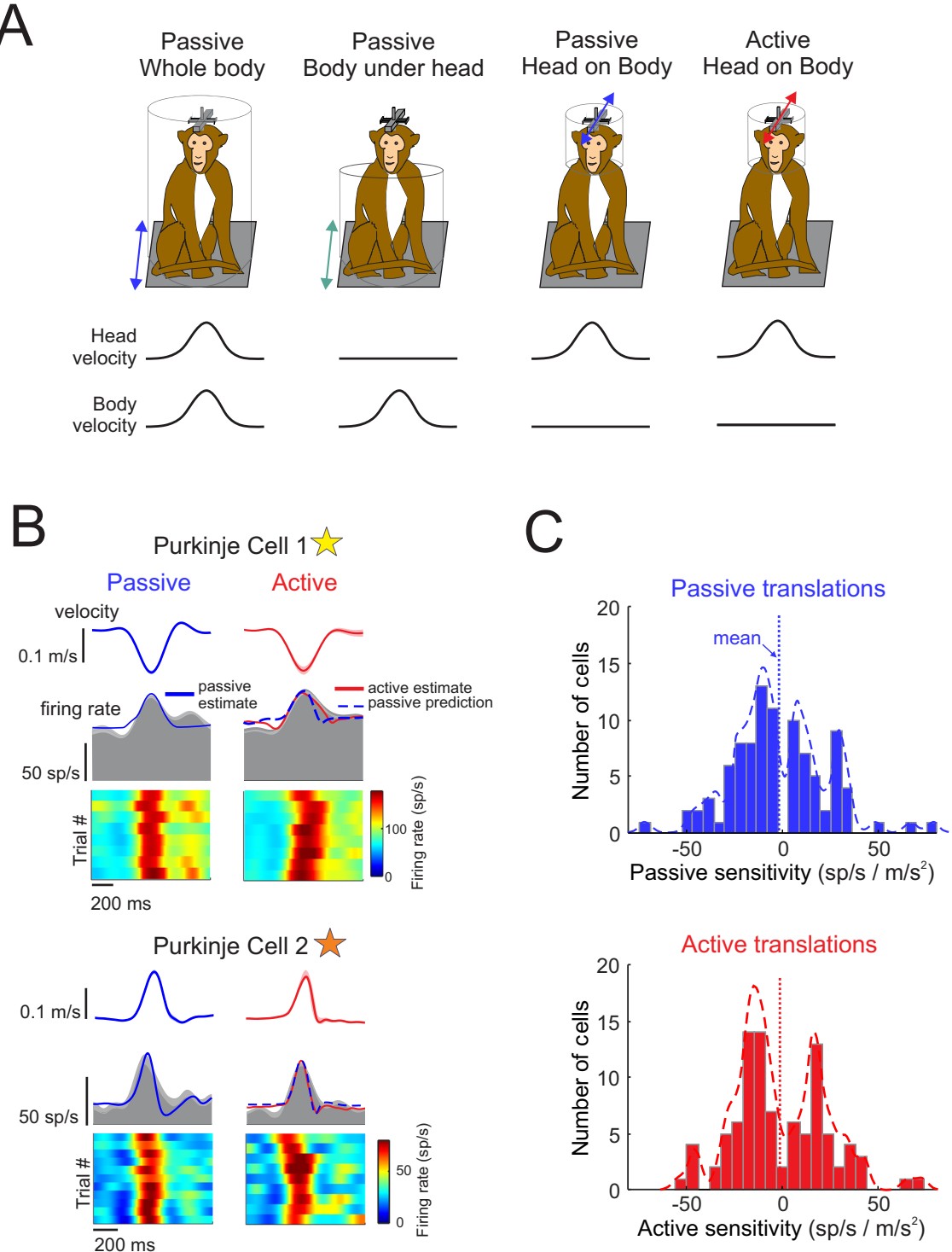

**Fig. 1 | Responses of Purkinje cell simple spikes to passively applied versus actively generated head translations. A** Schematic of paradigms to apply passive vestibular (whole-body translations), proprioceptive (body under head translations), and combined (head on body translations) stimulation, as well as allow the monkey to make active head translations. **B** The simple spike firing rate responses are shown for two example Purkinje cells during passive versus active head in body movement. Head on body velocity is shown in the top rows (shading represents ± standard error), and simple spike firing rate (gray shaded area) along with the linear estimation of firing rate based on head kinematics (superimposed blue and red solid) are shown in the bottom row. During active movement, the predicted firing rate based on the passive condition is superimposed (blue dashed line). Heat maps illustrate simple spike firing rates across all motion trials. **C** Distribution of Purkinje cell sensitivities to motion in the preferred direction (direction that generated the largest increase in firing rate) for active and passive conditions.

response (see Methods), we categorized tuning profiles as linear, rectifying, v-shaped, or other (example cells shown in Supplementary Fig. 4A), and found a comparable distribution of response types across conditions (Supplementary Fig. 4B). Unlike vestibular afferents and

nuclei neurons that encode motion linearly, Purkinje cells can exhibit heterogeneous and nonlinear tuning–a feature also reported in other regions of the vestibular vermis[4,22,23]. We then examined the specific temporal features driving each neuron's sensitivity – defined here as

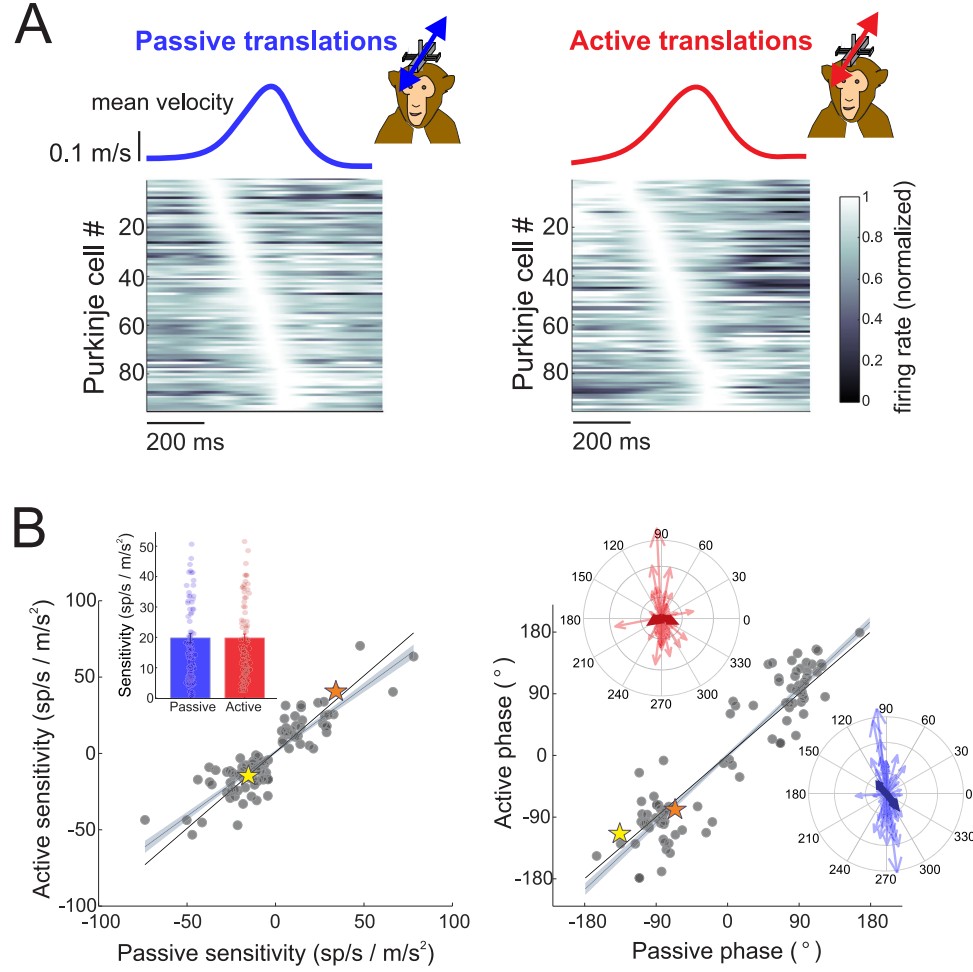

**Fig. 2 | Sensitivities and phases of all Purkinje cells to passive and active head motion. A** Firing rate responses across all Purkinje cells during passive and active head motion. Mean translation velocity is shown above, and below the firing rates for each Purkinje cell were normalized to peak firing rate and ordered based on the location of their peak firing rate for illustrative purposes. **B** Active versus passive Purkinje cell sensitivities and phases to head motion in the preferred movement direction ($N = 95$ neurons recorded from 2 animals). Gray line and shading represent the linear fit ±95% confidence interval. Stars indicate the sensitivities and phases of the two example Purkinje cells shown in Fig. 1. Insets: Bar graph demonstrating mean ± standard error passive versus active sensitivities, and polar plots showing the magnitude (vector length) and the phase (angle) of Purkinje cell responses to active (red) and passive (blue) head motion.

the depth of modulation – and found that response dynamics remained consistent across active and passive conditions. During passive head translations, 48% of NU Purkinje cells exhibited modulation most strongly aligned with linear acceleration, while 41% were velocity-aligned and 11% with jerk (Supplementary Fig. 4C, left panel). This heterogeneity persisted during active motion, with similar proportions of neurons aligned to acceleration (44%), velocity (44%), and jerk (13%) (Supplementary Fig. 4C, right panel). The persistence of this heterogeneity across movement contexts is also supported by the consistency in the phase as well as sensitivity of the response between active and passive conditions (Fig. 2B). These results reinforce the conclusion that the rich diversity in NU Purkinje cell response dynamics is not context-dependent, but rather reflects a stable and robust feature of cerebellar encoding across both self-generated and externally applied movements.

### NU Purkinje cells encode total self-motion in space when active and passive motion are experienced concurrently

So far, we have shown that NU Purkinje cells encode self-generated movement with similar fidelity as passively imposed movements. These results indicate that NU Purkinje cells provide a context-independent, ground truth representation of self-motion in space, whether

movements are actively generated or passively applied. We next asked whether this ground truth representation of self-motion in space is preserved when active and passive movements occur simultaneously. To test this, we applied passive whole-body anteroposterior translations while monkeys concurrently generated active head-on-body movements. Figure 3A illustrates the response of an example Purkinje cell during combined active and passive motion. The top traces show active head motion (red), passive whole-body translation (blue), and the resulting total head motion in space (purple), which is the sum of the two. The lower panel illustrates the neuron's firing rate (gray shaded region), with two model predictions overlaid: the blue dashed line reflects the prediction based on passive motion alone, while the solid purple line reflects the prediction based on total motion. Strikingly, the total-motion model well predicted the neuron's firing rate, whereas the passive-motion-based prediction failed to account for the response. Thus, this finding confirms that the neuron continues to encode an accurate, context-independent representation of self-motion in space, even when active and passive movements occur simultaneously.

The example in Fig. 3A illustrates that a single NU Purkinje cell integrates both active and passive components to represent total self-motion. To extend this observation across our population of NU Purkinje cells, we next conducted a quantitative model-based analysis. For

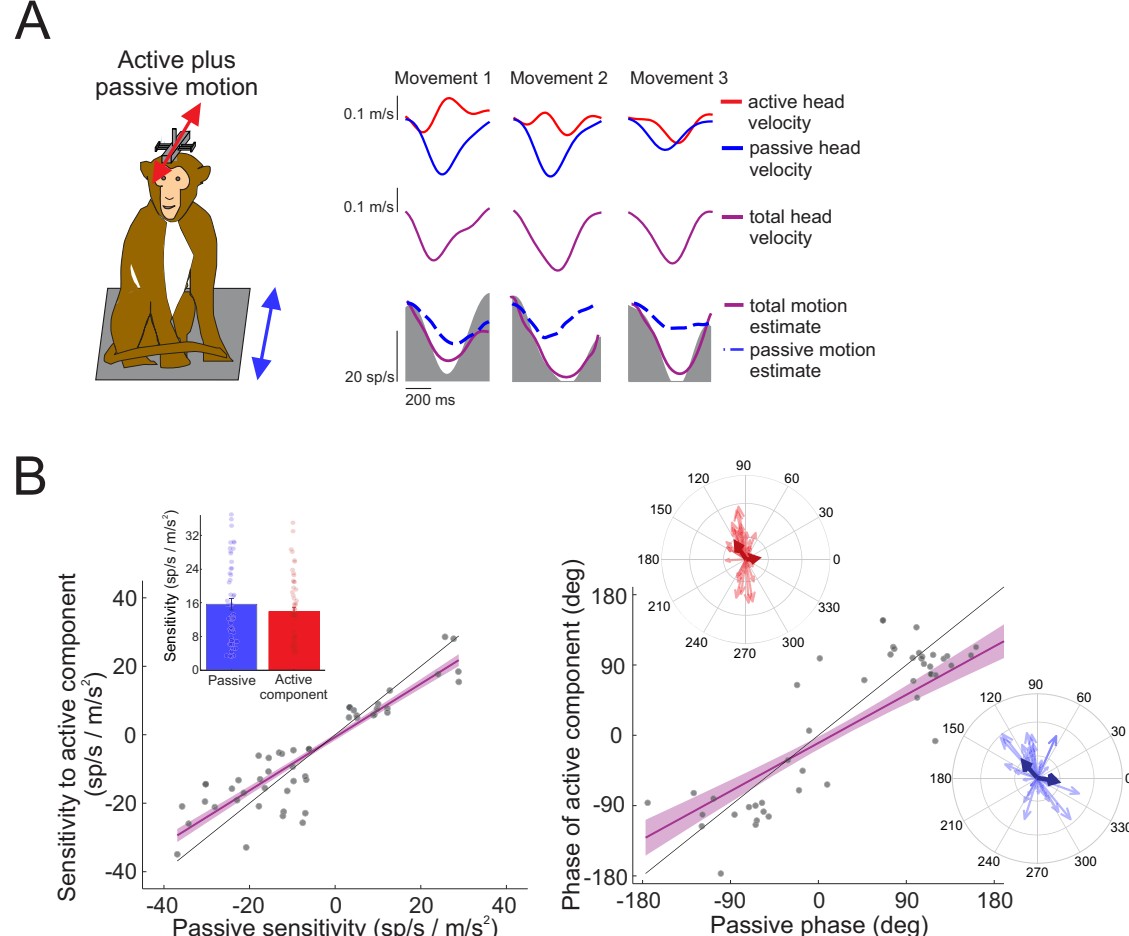

**Fig. 3 | Example Purkinje cell responses to combined active and passive motion. A** Passive motion stimulation was applied by translating the whole body while the monkey made active head movements. Active and passive components of head velocity are shown in red and blue traces on top, and total head velocity in space in purple below. Estimated firing rates based on passive motion only (blue dashed line) and total head motion (purple line) are superimposed on the actual neural firing rate (shaded gray). **B** Sensitivities and phases to the active versus

passive components of combined motion ($N$ = 52 neurons recorded from 2 animals). Purple line and shading represent the linear fit ±95% confidence interval. Insets: Bar graph demonstrates mean sensitivities ± standard error to the passive versus active components of head motion, and polar plots show the magnitude (vector length) and the phase (angle) of Purkinje cell responses to the active component (red) and passive (blue) head motion.

each neuron, we first estimated sensitivity to the passive component of motion generated by the applied passive whole-body translations. We then fit the residual firing rate – i.e., the portion not explained by passive motion – with the active component of head motion (see Methods). Across the 52 Purkinje cells tested, sensitivities to the active and passive components were statistically indistinguishable ($p$ = 0.5038). Furthermore, response phases to passive and active components were well aligned across neurons ($R^2$ = 0.5433, $p$ < 0.0001; Fig. 3B, right panel). Thus, taken together, these results establish that NU Purkinje cells provide a veridical representation of self-motion in space regardless of whether it occurs passively, is generated actively, or involves both in combination.

**Motor command signals do not influence NU Purkinje cell activity**

Our findings thus far demonstrate that NU Purkinje cells reliably encode self-motion in space by integrating vestibular and proprioceptive input, regardless of whether movement is actively generated, passively imposed, or experienced concurrently. This context-independent encoding suggests that these neurons are not influenced by motor command signals. To directly test this, we implemented a motor-only condition in which the monkey attempted to

initiate voluntary head movements, but actual motion was prevented. As a result, motor commands were issued to the neck musculature, but there was no head displacement in space (i.e., no vestibular input) and no head movement relative to the body (i.e., minimal proprioceptive feedback from neck receptors).

To confirm the presence of motor output, we recorded isometric force generation using a transducer mounted on the sled, capturing the onset of intended movement. Example responses from three Purkinje cells (the same cells shown in Fig. 1 plus an additional cell) are presented in Fig. 4A. In contrast to the robust modulation observed during both active and passive head motion, neither cell showed detectable modulation in activity when only motor commands were present. This result generalized across the population: in the 60 NU Purkinje cells tested, sensitivity during attempted head motion was significantly lower than during either active or passive movement ($p$ < 0.0001; Fig. 4A, bar graph). Furthermore, we assessed whether the baseline firing rates changed after motor commands were sent and found they remained stable between the 50 ms before and after the onset of intended movement ($p$ = 0.8090; Fig. 4B), consistent with the absence of efference-copy signals influencing NU Purkinje cell output.

Together, these findings demonstrate that NU Purkinje cell activity reflects only sensory input and is not influenced by motor

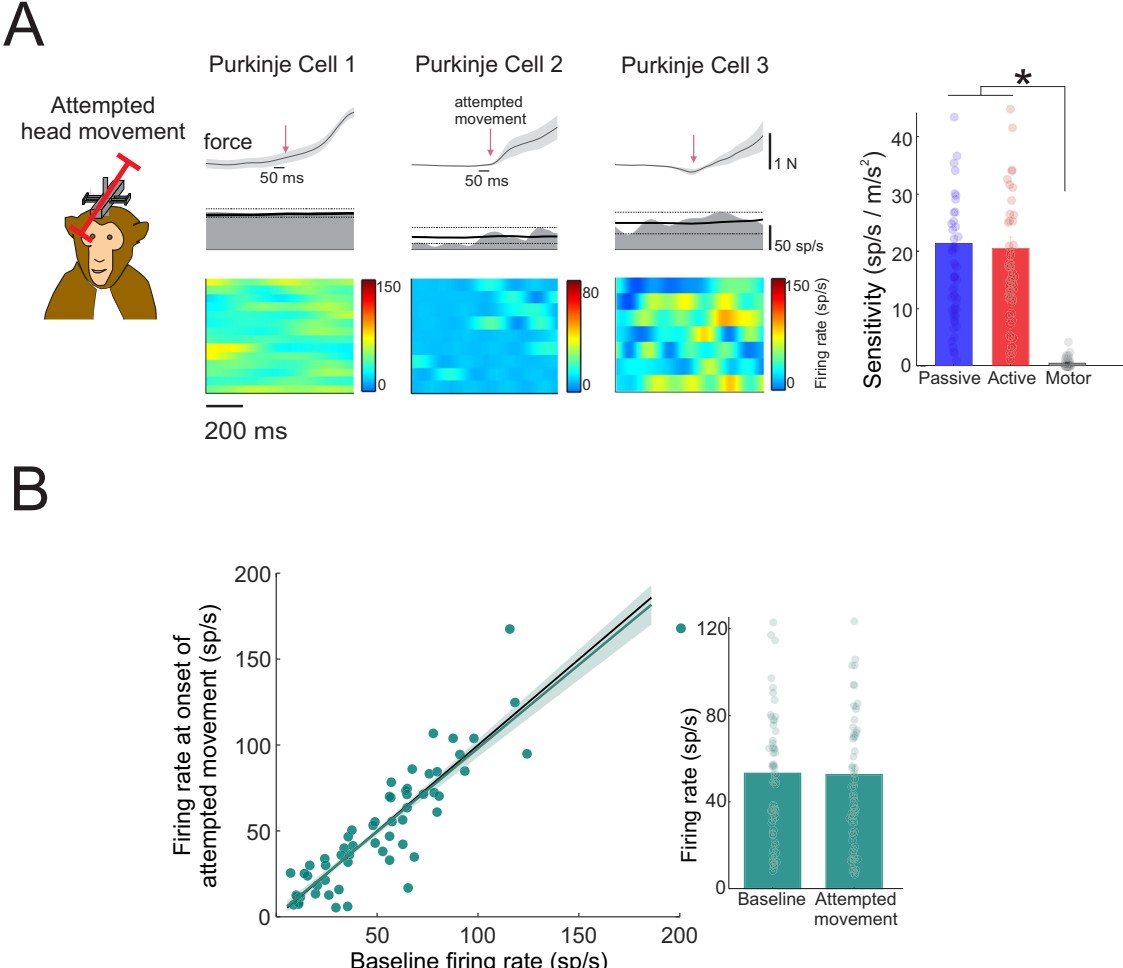

**Fig. 4 | Purkinje cell responses to motor command signals. A** Responses to motor commands were examined by restricting an intended head movement such that motor commands were sent to neck muscles, but no movement occurred. The force of the intended head movement, firing rate along with the mean ±2 standard deviations (black solid and dotted lines, respectively), and heat maps of firing rate across trials are shown for three example Purkinje cells (the same cells as shown in Fig. 1 as well as one additional Purkinje cell). Arrows indicate the onset of intended head movement. Bar graph demonstrates mean ± standard error of Purkinje cell sensitivities during the passive, active, and motor command conditions ($N = 60$ neurons recorded from 2 animals; *$p < 0.0001$, two-sided paired samples $t$ test; no adjustment for multiple comparisons). **B** Firing rate during 50 ms at the onset of the intended head movement vs. baseline firing rate. Gray line and shading represent the linear fit ±95% confidence interval. Bar graph demonstrates mean firing rate ± standard error at baseline versus the onset of the intended head movement ($N = 60$ neurons recorded from 2 animals).

commands. This sensory-specific tuning, coupled with the indistinguishable responses observed during active and passive motion, provides compelling evidence that NU Purkinje cells encode a context-independent, ground truth representation of self-motion in space. Notably, this pattern diverges from prior work in the anterior vermis, where Purkinje cells have been shown to respond to motor-related signals in the absence of actual movement. The functional implications of this regional distinction will be explored further in the Discussion.

### NU Purkinje cells encode active changes in head orientation relative to gravity

The NU receives direct input from both the otolith organs and semicircular canals and is a key site for integrating these signals. This integration is necessary because, as stated by Einstein's equivalence principle, the effects of gravity and linear acceleration are physically indistinguishable. The otoliths encode net linear acceleration and cannot, on their own, differentiate between head tilt relative to gravity and actual translation. Resolving this ambiguity requires combining otolith signals with rotational input from the semicircular canals. This computation has been demonstrated in the NU, with its output

observed in Purkinje cell activity during passive motion[24], reviewed in Cullen[11]. During head tilt, both the semicircular canals and otoliths are activated by angular rotation and by the change in head orientation relative to gravity, respectively, whereas linear translation stimulates only the otoliths. Importantly, recent work has shown that neurons in the fastigial nucleus, the downstream target of Purkinje cells in the vermis, can distinguish between active and passive tilts, consistent with the operation of an internal model that accounts for the gravitational component of otolith signals[25]. We therefore asked whether NU Purkinje cells, which provide input to this pathway, also differentiate active from passive tilts, or whether, as observed for translations, their encoding remains invariant across behavioral context.

To test whether NU Purkinje cells differentiate between active and passive head tilts, we first identified neurons that responded to passively applied head-on-body pitch rotations. We then recorded from the same neurons while the monkey performed active upward and downward head pitch movements for a reward. Figure 5A shows an example neuron's simple spike modulation during the dynamic phase of head reorientation relative to gravity in both conditions. This neuron exhibited highly similar firing patterns during active and passive

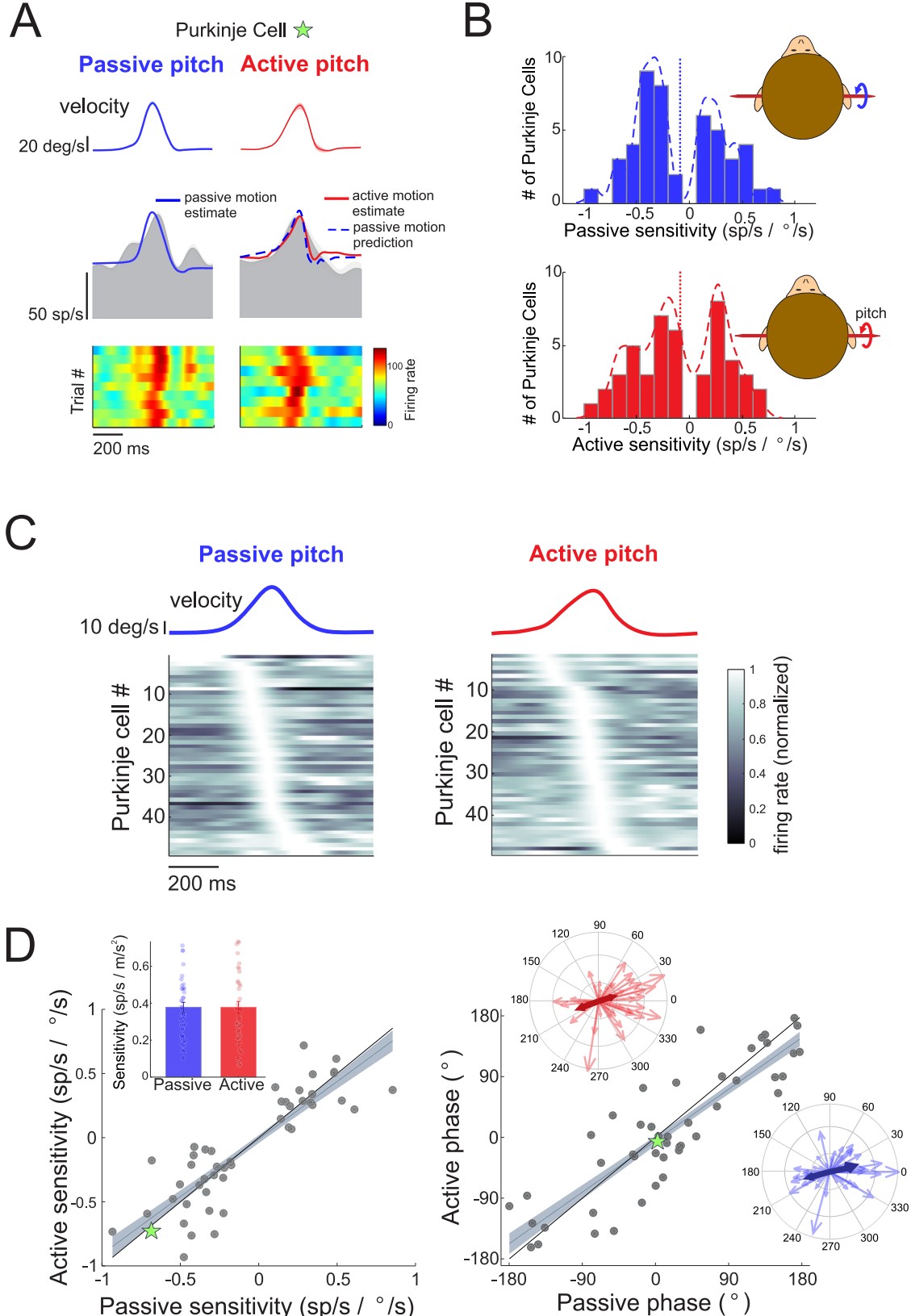

tilt. This similarity in response was robust across the population ($N = 47$), with nearly identical distributions of sensitivity to active and passive head tilts (Fig. 5B, histograms).

Furthermore, as shown in Fig. 5C, individual neurons displayed a broad range of firing phases during passive tilt (left), and this diversity was preserved during active tilt (right), collectively tiling the full cycle

of head motion. Importantly, each individual Purkinje cell maintained its tuning across conditions: sensitivity and phase were strongly correlated between active and passive tilts (Fig. 5D; sensitivity $R^2 = 0.7399$, phase $R^2 = 0.7627$), with no significant difference in sensitivity ($p = 0.9595$; bar graph inset). Comparable results were also observed in the non-preferred direction (sensitivity $R^2 = 0.7839$, phase $R^2 = 0.8357$;

**Fig. 5 | Responses of Purkinje cell simple spikes to passively applied versus actively generated dynamic head tilt. A** Passive motion stimulation was applied by tilting the head on the body in pitch. The simple spike firing rate responses are shown for one example Purkinje cell during passive versus similar actively generated head movement. Head velocity is shown in the top rows (shading represents ± standard error), and simple spike firing rate (gray shaded area) along with the linear estimation of firing rate based on head kinematics (superimposed blue and red solid) are shown in the bottom row. During active movement, the predicted firing rate based on the passive condition is also superimposed (blue dashed line). Heat maps illustrate simple spike firing rates across all motion trials. **B** Distribution of Purkinje cell sensitivities to motion in the preferred direction (direction that generated the largest increase in firing rate) for active and passive head tilts. **C** Firing rate responses across all Purkinje cells during passive and active head tilts. Firing rates for each Purkinje cell were normalized to peak firing rate and ordered based on the location of their peak firing rate for illustrative purposes. **D** Passive vs. active Purkinje cell sensitivities and phases to head tilt in the preferred direction ($N = 47$ neurons recorded from 2 animals). Gray line and shading represent the linear fit ±95% confidence interval. Star indicates the sensitivity and phase of the example Purkinje cell shown in panel (**A**). Insets: Bar graph demonstrates mean ± standard error passive versus active sensitivities, and polar plots showing the magnitude (vector length) and the phase (angle) of Purkinje cell responses to active and passive head tilt.

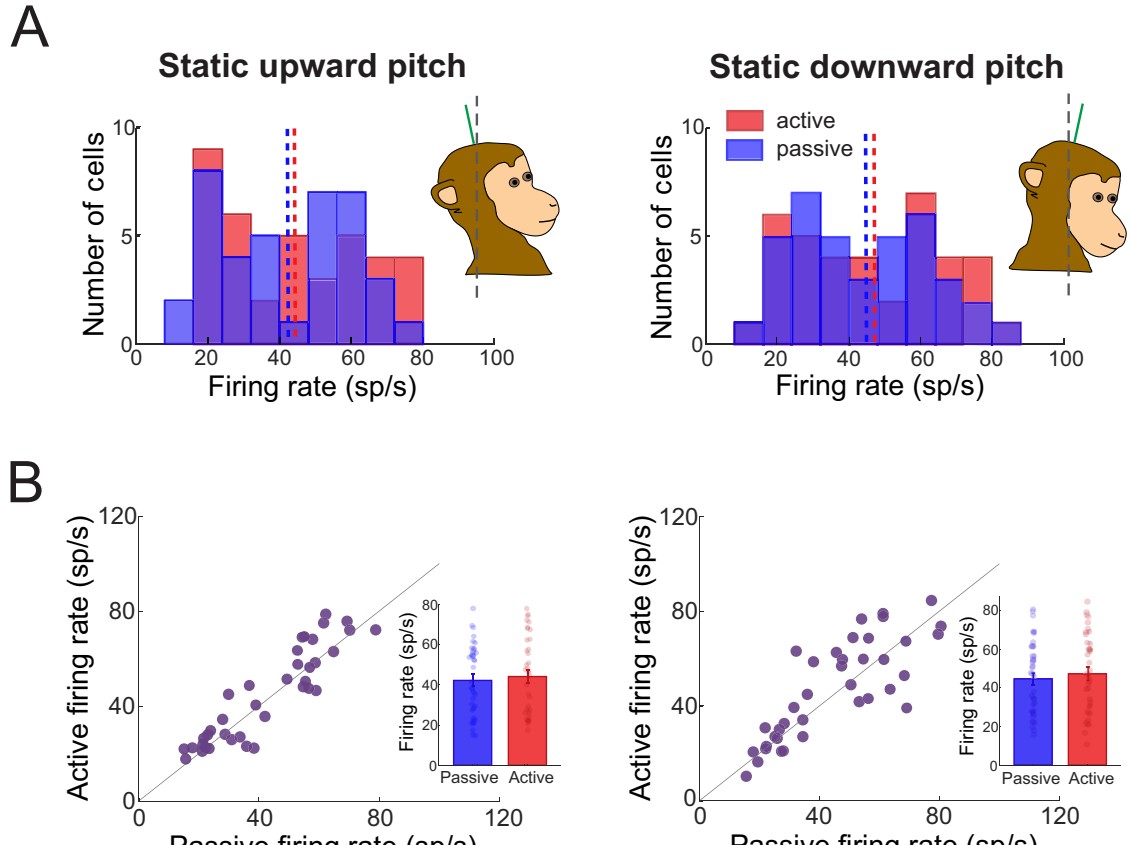

**Fig. 6 | Purkinje cell encoding of static changes in head orientation relative to gravity generated passively versus actively. A** Purkinje cell firing rate distributions across passive and active conditions during static upward (left panel) and downward (right panel) head pitch. **B** Passive vs. active individual Purkinje cell firing rates during static upward (left panel) and downward (right panel) head pitch. Black line represents the unity line. Bar graphs demonstrate mean ± standard error passive versus active sensitivities ($N = 38$ neurons recorded from 2 animals).

$p = 0.0775$; Supplementary Fig. 5). Thus, these findings reveal that, even in the context of gravitationally complex tilt movements, NU Purkinje cells preserve both the structure and fidelity of their responses across behavioral conditions – supporting context-invariant encoding of self-motion.

Finally, the analyses above focused on dynamic tilts, where Purkinje cell responses tracked changes in head orientation relative to gravity. Accordingly, we next asked whether NU neurons similarly encode static head position in a context-independent manner. Previous work has shown that gravity-related modulation in fastigial nucleus neurons is suppressed when the head is actively, rather than passively, brought into a fixed position[25]. To directly test whether this is also the case for NU Purkinje cells, we analyzed a subset of neurons ($N = 38$) recorded while the head was held motionless in an upward or downward pitch position that had been reached either by passively

tilting the head or through the animal's own voluntary movement. Overall, across our population, we found that the distribution of responses was comparable across both active and passive conditions (Fig. 6A). Furthermore and importantly, each individual Purkinje cell maintained its firing activity across conditions (Fig. 6B) (upward: $R^2 = 0.8269$; downward: $R^2 = 0.6598$; Fig. 6B), with no significant difference in sensitivity (upward: passive $42.4 \pm 18.0$ sp/s, active $44.4 \pm 19.6$ sp/s, $p = 0.1399$; downward: passive $44.6 \pm 19.1$ sp/s, active $47.2 \pm 20.7$ sp/s, $p = 0.2098$; bar graph inset).

Thus, in clear contrast to the fastigial nucleus, NU Purkinje cells maintain consistent encoding of orientation relative to gravity, regardless of whether it is achieved actively or passively–mirroring the context-independent encoding observed during dynamic translations and tilts. Together, these findings demonstrate that NU Purkinje cells

provide a stable, movement-invariant representation of orientation relative to gravity across both static and dynamic conditions.

## Discussion

Our findings reveal that Purkinje cells in the cerebellar nodulus and uvula (NU) encode a stable, veridical representation of self-motion and orientation relative to gravity, even during voluntary behavior. Contrary to internal model frameworks that posit suppression of predictable, self-generated input, NU responses to active head movements closely mirrored those elicited passively, with no evidence of reafferent cancellation. When active and passive motions were combined, Purkinje cell firing reflected the net motion in space, regardless of the source. Moreover, activity remained unchanged when motor commands were issued, but head movement was prevented, indicating no contribution from efference copy signals. During voluntary tilts, NU neurons reliably encoded both dynamic motion and static head orientation, recapitulating their responses during passive tilts. Together, these results challenge the assumption that predictive filtering is a universal principle of cerebellar processing and instead establish the NU as a context-invariant motion encoder. By providing a gravity-centered estimate of self-motion that remains stable across behaviors and motor contexts, the NU delivers a ground-truth reference frame against which internal predictions can be evaluated.

### Unambiguous coding of linear translations and head tilts relative to gravity in the NU

In striking contrast to other vestibular cerebellar regions (e.g., the anterior vermis and its downstream targets), NU Purkinje cells transmit unaltered signals of linear translations in space. Similarly, they encode active changes in orientation relative to gravity with a fidelity indistinguishable from their responses during passive tilts. These findings are particularly notable given the computational challenge of estimating gravitational orientation, which is not directly sensed but inferred by disambiguating otolith signals that reflect the net effect of head tilt and translation. Prior studies have shown that the NU integrates canal and otolith signals to resolve this ambiguity and compute an internal estimate of head orientation[26–31]. Based on this multisensory integration, a unified internal model framework has been proposed that the NU provides an optimal estimate of self-motion, formalized using a Kalman filter[12,32]. Central to this model is the use of motor signals to predict and suppress the sensory consequences of self-generated motion – thus allowing the residual (unpredicted) signals to be interpreted as externally imposed perturbations. Thus, our results overturn this core tenet of the model, revealing that NU Purkinje cells instead provide a veridical, context-invariant representation of self-motion and orientation relative to gravity.

To our knowledge, only one prior study has recorded Purkinje cell activity in the NU during active motion[33]. In contrast to our findings, Dugué et al.[33] reported that NU neurons in rats exhibited differential simple spike modulation during active head movements compared to passive whole-body motion, but several factors may account for this discrepancy. First, the active head movements in their study contained substantially higher frequencies than the passively applied whole-body motion. Because neural sensitivity can depend on the frequency content of the stimulus, comparing responses to stimuli with non-comparable frequency content can produce differences unrelated to context. Second, their passive condition consisted of whole-body rotation, which drives vestibular afferents but generates little or no proprioceptive input from the neck or body, whereas their active condition involved freely generated head movements that necessarily engage both vestibular and proprioceptive pathways. Thus, differences attributed to efference-copy mechanisms could instead reflect altered somatosensory input, given that NU Purkinje cells integrate proprioceptive cues[21]. Third, species differences may be critical: rats, as lateral-eyed quadrupeds, differ from foveate, upright primates in

cerebellar anatomy, postural demands, and the integration of vestibular and extravestibular inputs. Thus, divergent coding strategies likely reflect both biomechanical and ethological differences across taxa.

These considerations highlight a broader computational challenge: otolith afferents encode both linear acceleration and head tilt relative to gravity in an identical manner, rendering the signal inherently ambiguous[34,35]. While this problem has been widely studied under passive conditions, it becomes even more behaviorally relevant during active movement, when distinguishing reafferent from exafferent input is essential for motor control. For example, our prior studies in rhesus monkeys have shown that the responses of Purkinje cells in the anterior vermis are suppressed during active compared to passive movements[36]. At a population level, this suppression in turn contributes directly to the cancellation of reafferent vestibular input in downstream neurons of the deep cerebellar and vestibular nuclei. These target neurons mediate vestibulospinal reflexes and show reduced responses during active rotations[7,9,36], translations[8], static head tilts[25], and multidimensional head movements[37]. This gating mechanism prevents reflexive responses that would otherwise oppose voluntary movement through space[38,39].

In contrast, our findings demonstrate that the NU provides a stable, context-invariant representation of self-motion and gravitational orientation. Despite integrating egocentric proprioceptive and vestibular cues[21], NU output does not reflect motor commands or predictive signals. As a result, its activity signals the same estimate during both active and passive movements. This distinction has important implications for the role of the cerebellar vermis in supporting posture and homeostasis. In everyday life, self-motion arises from a mixture of voluntary actions and external perturbations. Reflex pathways – such as vestibulospinal circuits – must respond robustly to unexpected disturbances, yet flexibly adapt during intentional movement. Simultaneously, the nervous system must maintain an accurate estimate of self-motion and spatial orientation to stabilize the body and regulate autonomic function. For instance, antigravity muscle tone must be adjusted during voluntary forward leans, and cardiovascular and respiratory responses must be coordinated across postural transitions[40,41]. Our results suggest that the NU contributes to these vital processes by providing a ground-truth estimate of self-motion and orientation that is preserved across active and passive behavioral contexts. This unambiguous signal likely underpins a wide range of reflexive and homeostatic mechanisms essential for functional behavior[31,42–44].

### A dual-system architecture for context-dependent and context-invariant coding in the cerebellar vermis

Our findings reveal a mechanistic basis for functional specialization across the cerebellar vermis. NU Purkinje cells integrate proprioceptive input but are not influenced by motor commands–a distinction that underlies their invariant encoding of self-motion across behavioral contexts. In contrast, as noted above, Purkinje cells in the anterior vermis exhibit suppressed responses during active versus passive movements (Fig. 7A, bar graph)[6], consistent with the cancellation of reafferent signals in downstream vestibular pathways. This functional divergence likely arises from differences in afferent input: while both regions receive convergent vestibular and proprioceptive signals, only the anterior vermis encodes motor-related information (Fig. 7A, pink arrow). The absence of motor signals in the NU (Fig. 7B) thus provides a mechanistic explanation for its context-invariant output, offering a stable representation of self-motion and orientation relative to gravity.

Indeed, anatomical studies further support this dissociation. The NU receives direct proprioceptive input from the external cuneate nucleus, central cervical nucleus, and nucleus Z (Fig. 7B, green arrow)[45–47], and its Purkinje cells respond to neck muscle stimulation[48]. These neurons also receive climbing fiber input from specific

## A    Anterior vermis

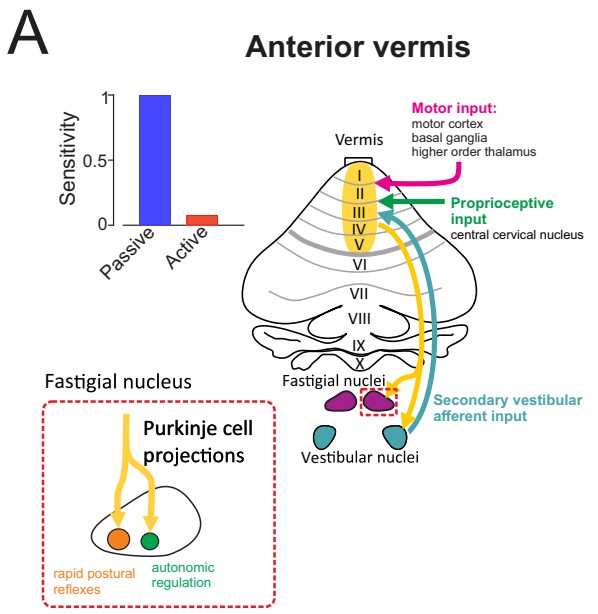

## B    Nodulus/Uvula

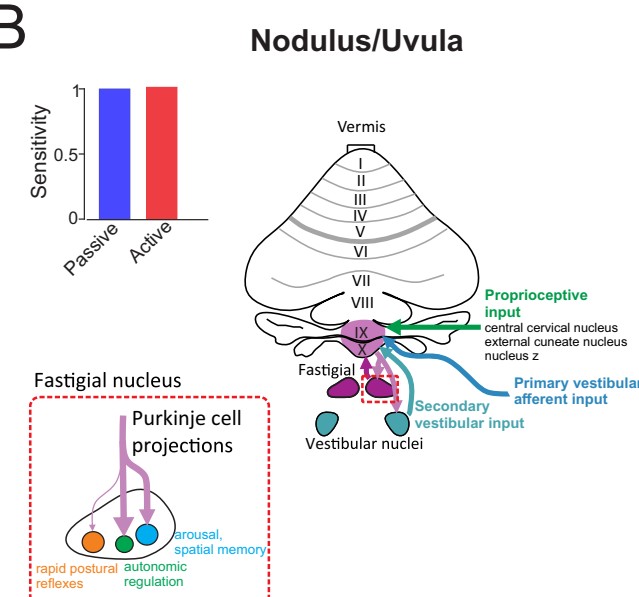

**Fig. 7 | Schematic comparison between the anterior and posterior vermis (nodulus/uvula) of sensorimotor inputs and projections. A** Schematic of the anterior vermis (shaded yellow) receiving motor, proprioceptive, and secondary vestibular input, and its Purkinje cell projections to cell groups in the fastigial nucleus involved in rapid postural reflexes and autonomic regulation. Bar graph demonstrates the distinction between encoding of passive versus active head

motion in the anterior vermis from Zobeiri and Cullen (2024). **B** Schematic of the nodulus/uvula (lobules X/IX; purple shading) receiving proprioceptive, and primary and secondary vestibular input, and its Purkinje cell projections to cell groups in the fastigial nucleus primarily involved in autonomic regulation, arousal, and higher order functions such as spatial memory. Bar graph demonstrates our observations of invariant encoding between passive versus active head motion.

subregions of the inferior olive and project to a ventral portion of the fastigial nucleus[17,49–51], which in turn targets subcortical and cortical regions involved in arousal, autonomic regulation, motor control, and higher-order functions[17–20]. The anterior vermis, by comparison, plays a more dynamic role in postural control. It receives mossy fiber input from the central cervical nucleus and secondary vestibular pathways[52], and its Purkinje cells respond to both vestibular and proprioceptive inputs[53–55]. These cells also receive climbing fiber projections from the beta subnucleus of the medial accessory olive and project to fastigial neurons that drive rapid postural reflexes[17]. Importantly, their sensitivity to motor commands[36] allows them to distinguish between active and passive motion, enabling suppression of predictable sensory input and refinement of reflex gain.

Together, these findings support the existence of two complementary functional streams within the cerebellar vermis. The anterior vermis enables flexible modulation of reflex pathways, selectively suppressing responses to self-generated motion to prevent interference with voluntary behavior. In parallel, the NU provides a veridical, context-invariant estimate of self-motion and gravitational orientation, which is critical for ongoing regulation of antigravity muscle tone and for coordinating autonomic adjustments during postural transitions. This dual-stream architecture ensures that the nervous system can both adaptively gate reflexes and maintain an accurate estimate of orientation in space–supporting behavioral stability, perception, and homeostasis.

### Broader implications for postural control and cerebellar dysfunction
While most prior work on the NU has focused on its role in gaze stabilization, particularly through analyses of the spatiotemporal dynamics of the (VOR)[26,27,54–57], it is important to recognize that NU function extends well beyond oculomotor control. Lesion studies in both humans and macaques have shown that, in addition to the well-documented oculomotor deficits, selective NU damage leads to

profound postural impairments, including head and trunk oscillations, impaired balance, and falls in all directions[58–62]. These behavioral impairments are consistent with a failure to accurately estimate body orientation and motion in space. Our findings provide a mechanistic explanation for these deficits by demonstrating that the NU integrates inputs from semicircular canals, otolith organs, and proprioceptors to compute a veridical, context-invariant estimate of head and body motion–even during voluntary movement. By elucidating the computational role of the NU in self-motion processing, this study advances our understanding of how the cerebellum supports posture and spatial orientation and offers critical insight into the neural mechanisms underlying balance disorders resulting from posterior cerebellar damage.

## Methods
### Experimental model and participant information
All experimental procedures performed in this study received full approval from the Johns Hopkins University Animal Care and Use Committee and adhered to the guidelines outlined by the United States National Institutes of Health (Protocol PR22M342). Neural recordings from the cerebellum were performed on one female and one male rhesus macaque (Macaca mulatta), weighing 8 and 11 kg. A total of 32 recording sessions were performed on separate days. These animals were maintained in a controlled environment with a 12-hour light/dark cycle throughout the duration of the study. Both rhesus monkeys exhibited overall good health, and did not require any medication during the course of this experiment, and had previously participated in other research studies within our laboratory.

### Surgical procedures
Each rhesus monkey underwent aseptic surgical preparations to allow for chronic extracellular recording following previously established protocols[63]. Each animal received pre-anesthetic medications, including ketamine hydrochloride (15 mg/kg i.m.), buprenorphine (0.01 mg/

kg i.m.), and diazepam (1 mg/kg i.m.), to ensure analgesia and muscle relaxation. To minimize swelling and prevent infection, loading doses of dexamethasone (1 mg/kg i.m.) and cefazolin (50 mg/kg i.v.) were administered. To stabilize heart rate and minimize salivation, anticholinergic glycopyrrolate (0.005 mg/kg i.m.) was also administered before surgery, and subsequently injected every 2.5–3 h during the surgical procedure. Anesthesia was maintained during surgery using isoflurane gas (0.8–1.5%), combined with a minimum of 3 l/min of 100% oxygen, adjusted to achieve the desired level of anesthesia. Vital signs, including heart rate, blood pressure, respiration, and body temperature, were monitored continuously throughout the surgical procedure.

Animals were implanted with a custom-made medical-grade titanium head post for head fixation and recording chambers that allowed targeting of the nodulus and uvula in the posterior cerebellar vermis. In the first rhesus monkey, stereotaxic targeting procedures were used to position the recording chambers, and targeting was examined post-implantation using a CT scan with guide tubes placed in the brain. In the second rhesus monkey, recording chambers were positioned based on the co-registration of a CT and MRI scan and by using Brainsight (Brainsight 2 Vet, Rogue Research, Montreal, Canada). A second CT scan was performed post-implantation and co-registered to the MRI to examine placement of the recording chambers and targeting of neural structures. Titanium screws and Simplex P bone cement (Stryker Orthopedics, Mahwah, NJ) were used to chronically secure the implant to the skull. Within the recording chamber, a craniotomy was carefully performed to provide access to the brain. Post-surgery, medications including dexamethasone (0.5 mg/kg i.m. for 4 days), anafen (2 mg/kg on the first day, 1 mg/kg on subsequent days), and buprenorphine (0.01 mg/kg i.m. every 12 h for 2–5 days, depending on the animal's pain level) were delivered. To prevent infection, Cefazolin (25 mg/kg) was injected twice daily for 10 days. Before beginning experimental procedures, the animals were allowed to recover for a 2 week period post-surgery.

## Experimental procedures

The monkey was head-fixed using a head post attached to a near-frictionless linear head sled that could enable translations of the head relative to the body in both the mediolateral and anteroposterior directions. The monkey was seated in a primate chair that was secured on top of a linear sled that could provide anteroposterior or mediolateral whole-body translations. Recordings were performed using acute electrode insertions on each session to record from a unique neural population. The electrode was inserted using a custom microdrive and a sterile guide tube that penetrated the tentorium. Electrodes were removed at the end of every session. During each recording, Purkinje cells were identified physiologically and by laminar context: as the electrode advanced from the molecular layer into the Purkinje cell layer, we observed complex spike–dominant activity in the dendritic region followed by simple spike activity at somatic depths, often still accompanied by identifiable complex spikes. The firing pattern of these simple spikes was tonic and irregular, consistent with Purkinje cell physiology. We first searched for and isolated neurons that showed modulation in response to vestibular stimulation and subsequently tested experimental conditions on these neurons. To test sensitivities to vestibular and proprioceptive stimulation, we applied passive vestibular stimulation by translating the whole body on the linear sled. We then held the head stationary relative to earth and applied passive proprioceptive stimuli by translating the body underneath a stationary head. We applied combined vestibular and proprioceptive stimulation by translating the head on an earth-stationary body. Finally, we allowed the monkey to make active voluntary translations in the anterior and posterior directions for a reward.

Combined active and passive motion was also examined by translating the whole body in the anterior and posterior direction at the same time that the monkey made an active head translation. Finally, in some trials, we unexpectedly locked the head sled such that motor commands were sent, but no head motion (and thus no sensory stimulation) occurred. In these trials, we recorded the force applied to the locked head sled by the monkey to determine when the onset of the intended head movement occurred. Similarly, to examine encoding of changes in head orientation relative to gravity, we first applied passive pitch and roll motion of the head on the body by releasing rotational and translation joints in the head post. Finally, we allowed the monkey to make active head pitch movements upwards and downwards for a reward.

## Data acquisition

We recorded 3D linear acceleration and 3D angular velocity of the head from an inertial measurement unit (IMU) mounted on the head post, linear motion of the primate chair using a 3D accelerometer, and the force applied to the head post using a load cell transducer (Omegadyne). Analogue signals were sampled at 1 kHz (Blackrock Microsystems). Neural activity was recorded from high-density (128ch) silicon read-write electrodes from the NU (IMEC). The 128 channels were arranged in a zig-zag pattern and spanned a 1.6 mm recording area. Neural signals were amplified, bandpass filtered (0.1 Hz–7.5 kHz) and digitized by four 32-channel RHS stim/recording head stages (Intan technologies) and streamed by an RHS stim/recording controller (Intan technologies, v3.4.0) at 30 kHz. Kilosort v2.5 was used to spike sort, and neural data were further curated in Phy2. Only isolated single units that showed significant modulation during passive vestibular stimulation ($p < 0.05$) and were present across a set of paradigms were included in further analyses.

## Quantification and analysis

Kinematic signals were low-pass filtered at 10 Hz, and linear velocity and jerk were calculated by integrating and differentiating the recorded acceleration signal, respectively, using MATLAB v9.8 (MathWorks). Similarly, angular position and acceleration signals were calculated by integrating and differentiating the recorded angular velocity signal from the IMU, respectively. Each trial of motion was extracted by segmenting the data at the onset and offset of movement. Firing rate was calculated using a kaiser filter with a cutoff of 7 Hz[64]. Each Purkinje cell's firing rate response to translational motion was estimated using linear regressions to find sensitivity to three kinematic terms (velocity, acceleration, and jerk).

$$\widehat{fr}(t) = b + S_v \dot{X}(t) + S_a \ddot{X}(t) + S_j \dddot{X}(t) \qquad (1)$$

where $\widehat{fr}(t)$ is the estimated firing rate, $b$ is a bias term representing background firing rate, $S_v$, $S_a$, and $S_j$ are coefficients representing the velocity, acceleration, and jerk gains respectively and $\dot{X}$, $\ddot{X}$ and $X$ are head velocity, acceleration, and jerk, respectively. For head-on-body motion (active or passive) we used kinematic signals recorded from the head. For body-under-head motion, we used kinematic signals recorded from the primate chair. The same analysis was performed to examine Purkinje cell sensitivity to active and passive head tilts; however, we used angular head position, velocity, and acceleration terms. This least-squares regression was solved for non-negative and non-positive criterion to ensure sign consistency across estimated coefficients. For each model coefficient in the analysis, we computed 95% confidence intervals using a nonparametric bootstrapping approach ($n = 2000$)[65,66]. All non-significant coefficients were set to zero. Similarly, we calculated a firing rate prediction for the active condition by substituting the gains computed for the passive condition into the equation above. During combined active and

passive motion, we first used kinematic signals from the chair, representing the passively applied motion, in the regression analysis to calculate sensitivity coefficients and generate a firing rate prediction. We then subtracted the passive estimate from the observed firing rate and fit the residual firing rate with the active component of motion, calculated as total head motion minus the applied chair motion. Coefficients were used to estimate the sensitivity and phase of each Purkinje cell's response using the following equations:

$$\text{Sensitivity} = \text{sgn}\left(S_j, S_a, S_{v,i}\right) \times \sqrt{\frac{\left((2\pi f)^2 S_j - S_v\right)^2 + (2\pi f S_a)^2}{(2\pi f)^2}} \quad (2)$$

$$\text{Phase} = \tan^{-1}\left(\frac{(2\pi f)^2 S_j - S_v}{2\pi f S_a}\right) \quad (3)$$

Where the sign terms were +1 or −1 for positive and negative coefficients, respectively, and the frequency ($f$) was 1 Hz to reflect the 500 ms half-cycle of motion that the stimulation consisted of. Similarly, sensitivities and phases for angular motion were also calculated, but instead using coefficients obtained for position, velocity, and acceleration.

We assessed the contribution of velocity, acceleration, and jerk to each Purkinje cell's response by running linear regressions without the term of interest and calculating the decrease in the variance accounted for (VAF). Purkinje cell responses to passive and active head-on-body motion were classified as linear, rectifying, or v-shaped. Linear neurons increased their firing rate in response to motion applied in one direction and decreased their firing rate with motion in the opposite direction, with a similar magnitude of sensitivity (within 8 sp/s/m/s²). Rectifying neurons primarily had responses only in one direction and minimal modulation in the other (below 8 sp/s/m/s²). V-shaped neurons showed similar increases in both motion directions (within 8 sp/s/m/s²). Remaining Purkinje cells that did not fit these three criteria were classified as other. We defined the preferred direction of the neuron as the direction of motion (e.g., anterior or posterior) that increased its firing rate, or had the larger increase in firing rate for v-shaped Purkinje cells.

Pairwise $t$ tests were used to compare absolute sensitivity to passive versus active head-on-body motion for both translations and the dynamic portion of head tilts. These comparisons were performed separately for the preferred and non-preferred movement directions. Pearson correlations were also used to assess the relationship between active versus passive sensitivity and phase, as well as between the passive versus the active component of combined motion for sensitivity and phase. During head pitch, we also extracted the static portion of head reorientations relative to gravity upwards and downwards for both passively applied movements and active movements. We examined the relationship between Purkinje cell firing rates during the static portion of the tilt between active and passive conditions Pearson correlations, and compared the firing rate between conditions using paired samples $t$ test (performed separately for upward and downward head pitch). Data are shown as mean ± SEM.

To examine Purkinje cell sensitivities to motor commands, we analyzed both the change in firing rate and the response to a theoretical head movement that would have occurred if the head was not restrained. Specifically, we compared Purkinje cell firing rates at the onset of movement (first 50 ms) relative to baseline (50 ms prior to the movement) using a paired samples $t$ test. We also superimposed the mean kinematics from the active head movements and calculated sensitivity to this theoretical motion, and compared this to passive and active head movement sensitivities.

## Reporting summary

Further information on research design is available in the Nature Portfolio Reporting Summary linked to this article.

## Data availability

The processed data that support the findings of this study are available in figshare with the identifier https://doi.org/10.6084/m9.figshare.30811046.

## Code availability

The custom written codes for this manuscript have been deposited on GitHub (https://github.com/CullenLab/Nodulus-uvula-encoding-of-motion).

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

## Acknowledgements

We thank Dale Roberts for providing technical assistance, Dr. Tim Harris for providing prototype high-density electrodes along with technical expertise, Lex Gomez for assistance with data processing, and all Cullen lab members for critical feedback on the manuscript and figures. This work was supported by funding from the National Institute on Deafness and Other Communication Disorders (R01-DC002390 and R01-DC018061) (K.E.C.), as well as a Natural Sciences and Engineering Postdoctoral Fellowship and Kavli Neuroscience Discovery Institute Distinguished Postdoctoral Fellowship (R.L.M).

## Author contributions

R.L.M. designed and conducted experiments, analyzed the data, and wrote the manuscript. K.E.C. designed the experiments, supervised the project, and wrote the manuscript.

## Competing interests

The authors declare no competing interests.
