## [Transparent Peer Review file · Nature Communications]

Ground-truth encoding of self-motion in the primate cerebellar nodulus and uvula

Corresponding Author: Dr Kathleen Cullen

Version 0:

Reviewer comments:

Reviewer #1

(Remarks to the Author)

The current study examined vestibular activity in the posterior part (NU) of the cerebellar vermis under active and passive self-motion condition. The main finding is that unlike many previously reported areas showing suppressed neural activity under active motion, NU activity remains, meaning not being suppressed by the motor command efference copy signal. This finding is important, because it provide a neural substrate that serves maintenance of monitoring self-motion perception from the vestibular signals during active motion. The experiments, including those controlled ones, are well performed, as always in their previous studies. The results are clear and persuasive. The paper is well written. I only have some minor comments for better presentations of their results.

1. line 23-24: The statement sounds too strong. It can be softened to something like: "Recent evidence suggest that cerebellar circuits in the anterior vermis may use efference copies..."
2. Vestibular and Proprioceptive, Supple 2B: show comparison of the two on a cell by cell basis, to see whether there is a correlation there; what about the preference? Are they consistent so that passive whole body rotation and neck proprioceptive response can be summed to enhance activity during passive head on body motion (their previous report)? Can show a similar scatter plot of comparison of preference between the two conditions.
3. Linear, rectifying, v-shaped neurons, Supple 4: show some examples for each category. In addition, it is a bit surprising that the linear group does not dominate, instead, there are quite a high proportion of rectifying and v-shaped neurons. Isn't that linear should be expected based on vestibular coding (excitation and inhibition mechanism, e.g. in peripheral)?
4. Line 313 : The same work here also involves proprioceptive stimulation (head on body during active and passive motion). So this is the same to the rat study. I am not sure about why it says it is a key distinction?
5. Data in many places are shown by pooling data from both animals. They need to be separated to indicate consistency across animals.
6. The population results are highly relying on fitting functions. Thus the goodness of the fit needs to show they are good fittings.
7. Figure 1C: why there are data missing in the middle around 0? Are there very few cells showing no modulation, or these cells are dropped out (with some criteria) in the presentation?
8. Figure 3b lacks representative polar plots comparable to those presented in Figure 2b.

Reviewer #2

(Remarks to the Author)

Nature Communications, Manuscript ID: NCOMMS-25-63595-T

General comments

This paper addresses one of the critical issues of neuroscience: How does the cerebellar cortex contribute to the so-called "cerebellar internal model"? The authors chose the cerebellar nodules and uvula (NU) (older parts of the cerebellar cortex that receive converging vestibular and somatosensory inputs). They examined the activities of NU cells under various external and internal conditions. They made three important observations: 1) NU cells responsive to passive head translations remained equally sensitive to self-generated movements; 2) these NU cells demonstrated a lack of efference-copy integration; 3) NU cells encoded both dynamic motion and static orientation relative to gravity. These findings overturn the internal model hypothesis and establish the NU as a ground-truth, context-invariant estimator of self-motion, supporting stable behavior in dynamic environments.

The experimental design, data analyses, conclusions, and discussion all appear reasonable. I have only some concerns about the identification of Purkinje cells and the presentation of the data.

Specific comments

1. Page 5, Results, line 64, "Purkinje cells": I did not find an identification of PCs in either the Results or Methods sections. Conventionally, the identification of PCs depends on the non-simultaneous co-existence of simple spikes and complex spikes in the same unit. Of course, I understand your recording system may not allow its application for all neurons. Nevertheless, the authors should describe the identification of PCs in detail, as it is crucial for addressing the question raised in this study.
2. If the recording electrodes were implanted chronically in the cerebellar cortex, how is it possible to distinguish neurons recorded from the same electrode on different recording sessions?
3. The data presented on pages 6 and 7 are impressive and convincing.
4. Page 7, lines 139-141, "Using a model-based decomposition of each neuron's response (see 140 Methods), we categorized tuning profiles as linear, rectifying, v-shaped, or other, and found a comparable distribution of response types across conditions (Extended Data Fig. 4A)": It is helpful to show an example of activity for each category of neurons.
5. Page 7, lines 146-148, "This heterogeneity persisted during active motion, with similar proportions of neurons aligned to acceleration (44%), velocity (44%), and jerk (13%) (Extended Data Fig. 4B, right panel)": Did you find the persistence of the heterogeneity for individual neurons, or did you see changes in tuning profiles between different movement contexts?
6. Page 9, lines 204-205, "Furthermore, firing rates did not differ between the 50 ms before and after the onset of intended movement ($p = 0.8090$; Fig. 4B)": The two panels in Fig. 4B are not consistent with each other. In the left panel, the firing rate of PCs shows significant variance for the baseline condition and the movement onset, while in the bar graph, the variance appears minimal.
7. Page 17, line 471, "IMU": What is IMU? I did not find its explanation in the manuscript.
8. Page 26, Figure 1C: I recommend converting the upper and lower panels into a 2D plot to illustrate the relationship between PC sensitivities for the active and passive conditions.
9. Page 28, Figure 2B: For both sensitivity and phase, PCs tend to make two clusters: one positive and the other negative. It is helpful to discuss its interpretations.
10. Page 29, Figure 3A: Does the difference between the purple line (total motion estimate) and the blue-hatched line (passive motion estimate) represent an active motion estimate?
11. Page 29, Figure 3B: The inset bar graph in the left panel: How did you calculate the mean sensitivities? There is a discrepancy between the significant variance in the 2-D plot and the small variance in the bar graph.
12. Page 30, Figure 4A: The same two neurons are used in both Figs. 1 and 4. Are they truly representative of the whole population of PCs?
13. Page 30, Figure 4A: There is no explanation about the bar graph in the Figure legend.
14. Page 30, Figure 4B: It is helpful to show the 50 ms time window for Fig. 4B in Fig. 4A.
15. Page 30, Figure 4B, the bar graph: It should demonstrate modulation from the baseline, rather than the raw firing rate.
16. Page 34, Figure 7: It is helpful to show the references in the legend.

Version 1:

Reviewer comments:

Reviewer #1

(Remarks to the Author)

The authors have addressed my comments well. I do not have further questions.

Reviewer #2

(Remarks to the Author)

I really appreciate the authors' efforts to prepare replies for my comments. I am convinced with the revised manuscript.

REVIEWER COMMENTS

Reviewer #1 (Remarks to the Author):

The current study examined vestibular activity in the posterior part (NU) of the cerebellar vermis under active and passive self-motion condition. The main finding is that unlike many previously reported areas showing suppressed neural activity under active motion, NU activity remains, meaning not being suppressed by the motor command efference copy signal. This finding is important, because it provide a neural substrate that serves maintenance of monitoring self-motion perception from the vestibular signals during active motion. The experiments, including those controlled ones, are well performed, as always in their previous studies. The results are clear and persuasive. The paper is well written. I only have some minor comments for better presentations of their results.

We thank the reviewer for the positive feedback on our manuscript regarding our well performed experiments and persuasive results. As detailed below, we have addressed each of the reviewer's comments.

1. line 23-24: The statement sounds too strong. It can be softened to something like: "Recent evidence suggest that cerebellar circuits in the anterior vermis may use efference copies..."

We have revised this sentence to: "*Recent evidence suggests cerebellar circuits in the anterior vermis use efference copies of motor commands to build forward internal models that predict the sensory consequences of behavior*⁶."

2. Vestibular and Proprioceptive, Supple 2B: show comparison of the two on a cell by cell basis, to see whether there is a correlation there; what about the preference? Are they consistent so that passive whole body rotation and neck proprioceptive response can be summed to enhance activity during passive head on body motion (their previous report)? Can show a similar scatter plot of comparison of preference between the two conditions.

We have added histograms of sensitivities in the non-preferred direction (Supplementary Fig. 2B). We have also added scatter plots showing that combined stimulation enhances responses relative to vestibular stimulation alone (points falling above unity), as well as bar graphs comparing responses for vestibular, proprioceptive and combined conditions.

3. Linear, rectifying, v-shaped neurons, Supple 4: show some examples for each category. In addition, it is a bit surprising that the linear group does not dominate, instead, there are quite a high proportion of rectifying and v-shaped neurons. Isn't that linear should be expected based on vestibular coding (excitation and inhibition mechanism, e.g. in peripheral)?

We appreciate the reviewer's comment and have added representative examples of linear, rectifying, and v-shaped Purkinje cell response profiles to Supplementary Fig. 4A.

Regarding the distribution of response types, while vestibular primary afferents and vestibular nuclei neurons typically display linear encoding, Purkinje cell simple-spike responses often do not. This nonlinearity likely emerges from multiple stages of processing within the cerebellar cortex. Vestibular and somatosensory afferents undergo substantial transformation as they pass through the granule-cell layer, where convergence of multiple inputs onto individual granule cells can introduce nonlinearities. In addition, mossy fiber and parallel fiber synapses exhibit a range of short-term plasticity properties, including synaptic depression and facilitation, which further shape Purkinje cell responses (Chabrol et al., 2015).

In this context, we have revised the Results (Lines 147–149) to note that nonlinear tuning patterns have been documented previously in other regions of the vestibular cerebellum, including the anterior vermis (Zobeiri & Cullen, 2022, 2024, Mildren and Cullen 2025), and in Purkinje cell responses during locomotion (Sauerbrei et al., 2015). Thus, while linear encoding is common in earlier stages of vestibular processing, nonlinear response profiles in Purkinje cells are well established and expected.

4. Line 313 : The same work here also involves proprioceptive stimulation (head on body during active and passive motion). So this is the same to the rat study. I am not sure about why it says it is a key distinction?

We appreciate the reviewer's comment and have revised Lines 323–332 to explicitly state that the rat study conflated two variables: active movement and proprioceptive input. Their passive condition consisted of whole-body rotation, which drives vestibular afferents but generates little or no proprioceptive input from the neck or body, whereas their active condition involved freely generated head movements that necessarily engage both vestibular and proprioceptive pathways. Thus, differences attributed to efference-copy mechanisms could instead reflect altered somatosensory input, given that NU Purkinje cells integrate proprioceptive cues.

In contrast, our study directly compares active head-on-body motion to passively applied head-on-body motion, isolating the effect of motor command while holding neck-proprioceptive input constant. This distinction is important, and we have revised the text to more explicitly emphasize it.

5. Data in many places are shown by pooling data from both animals. They need to be separated to indicate consistency across animals.

To demonstrate consistency across animals, we now include separate data for each monkey in Supplementary Fig. 3C. These panels clearly show that the key active–passive comparisons hold in both subjects.

6. The population results are highly relying on fitting functions. Thus the goodness of the fit needs to show they are good fittings.

We have examined the goodness of fit using variance accounted for (VAF) and included this information in the results (lines 99-101). We found the fits are robust and the

average VAFs were comparable in the passive and active conditions (0.43 ± 0.18 and 0.41 ± 0.17 , respectively ($p = 0.5$).

7. Figure 1C: why there are data missing in the middle around 0? Are there very few cells showing no modulation, or these cells are dropped out (with some criteria) in the presentation?

We thank the reviewer for highlighting this point. We have expanded the Methods (Lines 478–479 and 507–508) to explain that we deliberately included only vestibular-sensitive neurons in our sample. Neurons with negligible vestibular modulation were excluded during both recording and post-processing. Therefore, the distribution naturally contains fewer points near zero.

8. Figure 3b lacks representative polar plots comparable to those presented in Figure 2b.

We have added polar plots to Figure 3B, consistent with the presentation format in previous figures.

Reviewer #2 (Remarks to the Author):

Nature Communications, Manuscript ID: NCOMMS-25-63595-T

Title: Ground-truth encoding of self-motion in the primate cerebellar nodulus and uvula.

Authors: Mildren RL, Cullen KE

General comments

This paper addresses one of the critical issues of neuroscience: How does the cerebellar cortex contribute to the so-called “cerebellar internal model”? The authors chose the cerebellar nodules and uvula (NU) (older parts of the cerebellar cortex that receive converging vestibular and somatosensory inputs). They examined the activities of NU cells under various external and internal conditions. They made three important observations: 1) NU cells responsive to passive head translations remained equally sensitive to self-generated movements; 2) these NU cells demonstrated a lack of efference-copy integration; 3) NU cells encoded both dynamic motion and static orientation relative to gravity. These findings overturn the internal model hypothesis and establish the NU as a ground-truth, context-invariant estimator of self-motion, supporting stable behavior in dynamic environments.

The experimental design, data analyses, conclusions, and discussion all appear reasonable. I have only some concerns about the identification of Purkinje cells and the presentation of the data.

We thank the reviewer for their thoughtful comments and address each point below.

Specific comments

1. Page 5, Results, line 64, “Purkinje cells”: I did not find an identification of PCs in either the Results or Methods sections. Conventionally, the identification of PCs depends on the non-simultaneous co-existence of simple spikes and complex spikes in the same unit. Of course, I

understand your recording system may not allow its application for all neurons. Nevertheless, the authors should describe the identification of PCs in detail, as it is crucial for addressing the question raised in this study.

We have revised the manuscript to provide a clearer description of the identification criteria. Specifically, we have now expanded the Methods (Lines 473-477) to explain that advancing the electrode from the molecular layer into the Purkinje cell layer produced initially complex spike activity in the dendritic region, followed by simple spike activity (sometimes with identifiable complex spikes as well) at the Purkinje cell layer. These recordings had a depth consistent with the PC layer (preceded by complex spike activity, and above clearly discernible nuclei neuron activity), as well as typical tonic, irregular simple spike firing typical of Purkinje cells.

2. If the recording electrodes were implanted chronically in the cerebellar cortex, how is it possible to distinguish neurons recorded from the same electrode on different recording sessions?

We clarified in the Methods (Lines 470–473) that the electrodes were acutely inserted during each recording session, withdrawn afterward, and reinserted at new sites for subsequent sessions. Thus, each session sampled unique neural populations.

3. The data presented on pages 6 and 7 are impressive and convincing.

We thank the reviewer for this encouraging feedback!

4. Page 7, lines 139-141, “Using a model-based decomposition of each neuron's response (see 140 Methods), we categorized tuning profiles as linear, rectifying, v-shaped, or other, and found a comparable distribution of response types across conditions (Extended Data Fig. 4A)”: It is helpful to show an example of activity for each category of neurons.

We have added examples of a linear, rectifying, and v-shaped neurons to Supplementary Fig. 4A.

5. Page 7, lines 146-148, “This heterogeneity persisted during active motion, with similar proportions of neurons aligned to acceleration (44%), velocity (44%), and jerk (13%) (Extended Data Fig. 4B, right panel)”: Did you find the persistence of the heterogeneity for individual neurons, or did you see changes in tuning profiles between different movement contexts?

We observed a persistence of this heterogeneity across movement contexts for individual neurons. This finding is also supported by the consistency in the phase as well as sensitivity of the response between active and passive conditions in Fig 2B. We have clarified this point in the text (lines 156-158).

6. Page 9, lines 204-205, “Furthermore, firing rates did not differ between the 50 ms before and after the onset of intended movement ($p = 0.8090$; Fig. 4B)”: The two panels in Fig. 4B are not

consistent with each other. In the left panel, the firing rate of PCs shows significant variance for the baseline condition and the movement onset, while in the bar graph, the variance appears minimal.

The left panel displays per-neuron values (where there is substantial between-cell variability as the reviewer noted). In contrast, the bar plot summarizes the population mean \pm SEM across neurons. With a larger sample, SEM is small even when between-cell variance is wide, which is why the bar appears tighter. Our paired comparison of the 50-ms pre- vs post-onset windows showed no systematic shift in firing rate ($p = 0.809$), consistent with the bar plot. To improve clarity, we have added mean \pm standard error in the figure captions.

7. Page 17, line 471, “IMU”: What is IMU? I did not find its explanation in the manuscript.

We have defined the acronym for IMU (Inertial measurement unit) where it is first mentioned.

8. Page 26, Figure 1C: I recommend converting the upper and lower panels into a 2D plot to illustrate the relationship between PC sensitivities for the active and passive conditions.

We thank the reviewer for this suggestion and examined it by constructing a version of the histogram that illustrates the relationship between PC sensitivities in active and passive conditions (see plot). However, we found that it did not provide additional insight into our findings given that the relationship between active and passive sensitivities is already shown directly and explicitly in the neuron-by-neuron scatter plots in Fig. 2B.

These plots display the same values the reviewer is asking to visualize and do so more clearly than a multidimensional histogram. As the additional plot would therefore be redundant, we have elected not to include it in the manuscript.

9. Page 28, Figure 2B: For both sensitivity and phase, PCs tend to make two clusters: one positive and the other negative. It is helpful to discuss its interpretations.

We have revised the text to explain that neurons with positive sensitivities prefer motion in the posterior direction, whereas those with negative sensitivities prefer anterior motion, and sensitivities tend to cluster around a similar absolute value regardless of whether posterior or anterior motion is preferred. Correspondingly, phases tended to cluster around a similar lead or lag relative to acceleration. These specific additions are now included in the revised Results section (Lines 133–137).

10. Page 29, Figure 3A: Does the difference between the purple line (total motion estimate) and the blue-hatched line (passive motion estimate) represent an active motion estimate?

We have revised the Methods (lines 535-536) to more clearly explain that the active component represents the residual after subtracting the passive predicted firing rate from the observed rate. Thus, the difference between the firing rate (which is similar to the total motion estimate) and passive estimate should roughly match with the active estimate.

11. Page 29, Figure 3B: The inset bar graph in the left panel: How did you calculate the mean sensitivities? There is a discrepancy between the significant variance in the 2-D plot and the small variance in the bar graph.

We now specify in the Methods that sensitivities are expressed as absolute values (to combine positive and negative frequencies) and that error bars represent mean \pm SEM (see also response to comment 6 above).

12. Page 30, Figure 4A: The same two neurons are used in both Figs. 1 and 4. Are they truly representative of the whole population of PCs?

We appreciate this comment and agree it is important to demonstrate that the example neurons are representative. We intentionally illustrate the responses of the same cells across figures to avoid the risk of “cherry picking” different examples for different conditions/analyses. To further address the reviewer’s concern, we now indicate the location of these specific cells in the population data shown in the heat maps in Figures 2A (translations) and 5C (pitch). Additionally, we note that the lack of responses to motor command signals is very consistent, as quantified for the population in Figure 4B, and we have added an additional Purkinje cell to Figure 4A to illustrate this point.

13. Page 30, Figure 4A: There is no explanation about the bar graph in the Figure legend.

We thank the reviewer for noting this and have added reference to the bar graph in the figure legend.

14. Page 30, Figure 4B: It is helpful to show the 50 ms time window for Fig. 4B in Fig. 4A.

We agree and have added a horizontal bar to indicate the 50 ms pre-movement window and an arrow to indicate movement onset in Fig. 4A.

15. Page 30, Figure 4B, the bar graph: It should demonstrate modulation from the baseline, rather than the raw firing rate.

We appreciate the reviewer’s comment and have revised the Results text (lines 214-218) to improve clarity regarding the analysis. Notably, panel A shows whether example Purkinje cells modulate their activity during attempted head movement, while Panel B evaluates whether firing rate changes systematically when motor commands are issued. Using raw firing rates for this comparison is important because subtracting baseline would artificially introduce variability (as any noise in the two windows would create a non-zero difference). That difference is already

captured by the distance from the unity line in the scatter plot of Fig. 4B. The raw firing rate comparison in the bar graph demonstrates that firing rates remain stable across the pre- and post-onset windows, consistent with the absence of efference-copy signals influencing NU Purkinje cell output.

16. Page 34, Figure 7: It is helpful to show the references in the legend.

Thank you — we are happy that the included reference was helpful.

REVIEWERS' COMMENTS

Reviewer #1 (Remarks to the Author):

The authors have addressed my comments well. I do not have further questions.

We thank the reviewer for their thorough evaluation and valuable feedback, which helped strengthen our manuscript.

Reviewer #2 (Remarks to the Author):

I really appreciate the authors' efforts to prepare replies for my comments. I am convinced with the revised manuscript.

We appreciate the reviewer's careful consideration and constructive comments throughout the review process.